# Iron, Heme Synthesis and Erythropoietic Porphyrias: A Complex Interplay

**DOI:** 10.3390/metabo11120798

**Published:** 2021-11-23

**Authors:** Antoine Poli, Caroline Schmitt, Boualem Moulouel, Arienne Mirmiran, Hervé Puy, Thibaud Lefèbvre, Laurent Gouya

**Affiliations:** 1Centre de Recherche sur L’Inflammation, Université de Paris, INSERM UMR 1149, 16 Rue Henri Huchard, 75018 Paris, France; caroline.schmitt@aphp.fr (C.S.); herve.puy@aphp.fr (H.P.); thibaud.lefebvre@aphp.fr (T.L.); laurent.gouya@inserm.fr (L.G.); 2AP-HP, Centre Français des Porphyries, Hôpital Louis Mourier, 178 Rue des Renouillers, 92700 Colombes, France; boualem.moulouel@aphp.fr (B.M.); arienne.mirmiran@aphp.fr (A.M.)

**Keywords:** erythropoietic protoporphyria, congenital erythropoietic porphyria, ALAS2, ferrochelatase, iron, iron-sulfur cluster, protoporphyrin IX, hematopoiesis

## Abstract

Erythropoietic porphyrias are caused by enzymatic dysfunctions in the heme biosynthetic pathway, resulting in porphyrins accumulation in red blood cells. The porphyrins deposition in tissues, including the skin, leads to photosensitivity that is present in all erythropoietic porphyrias. In the bone marrow, heme synthesis is mainly controlled by intracellular labile iron by post-transcriptional regulation: translation of *ALAS2* mRNA, the first and rate-limiting enzyme of the pathway, is inhibited when iron availability is low. Moreover, it has been shown that the expression of ferrochelatase (FECH, an iron-sulfur cluster enzyme that inserts iron into protoporphyrin IX to form heme), is regulated by intracellular iron level. Accordingly, there is accumulating evidence that iron status can mitigate disease expression in patients with erythropoietic porphyrias. This article will review the available clinical data on how iron status can modify the symptoms of erythropoietic porphyrias. We will then review the modulation of heme biosynthesis pathway by iron availability in the erythron and its role in erythropoietic porphyrias physiopathology. Finally, we will summarize what is known of FECH interactions with other proteins involved in iron metabolism in the mitochondria.

## 1. Introduction

Erythropoietic porphyrias are inborn errors of heme biosynthesis resulting from the altered activity of an enzyme in the pathway and leading to the primary accumulation of porphyrins in the erythron (Figure 1) [1]. The porphyrins deposition in tissues is responsible for the cutaneous photosensitivity of patients. Anemia is often present. The erythropoietic porphyrias mainly encompass two distinct diseases.

Firstly, congenital erythropoietic porphyria (CEP) is most often caused by a deficiency in uroporphyrinogen III synthase (UROS), the fourth enzyme of the heme biosynthesis pathway [2]. It results in accumulation of uroporphyrin I and coproporphyrin I in the red blood cells that cannot be further metabolized. CEP has a broad spectrum of phenotypic manifestations ranging from hydrops fetalis to late and mild cutaneous involvement [3]. The most frequent hematological finding is a hemolytic anemia that can be transfusion dependent in severe cases. Of note, few patients were found to harbor mutations in *GATA1*, on the X chromosome [4].

Secondly, erythropoietic protoporphyrias that can be subdivided into two types depending on the gene alteration responsible for the disease. Classical erythropoietic protoporphyria (EPP) is characterized by reduced activity of ferrochelatase (FECH), the last enzyme of the pathway [1]. It catalyzes the insertion of iron into protoporphyrin IX (PPIX) to form heme. In most patients, EPP is caused by a rare loss-of-function mutation in *FECH* associated in trans to a frequent hypomorphic variant which creates an aberrant splice site [5]. FECH partial deficiency leads to free PPIX accumulation, responsible for the phototoxic reactions experienced by EPP patients. Gain-of-function mutations in *ALAS2*, the first enzyme of erythroid heme synthesis, cause X-linked dominant protoporphyria (XLPP) in approximatively 4–10% of the patients with erythropoietic protoporphyria [6,7]. XLPP is clinically similar to EPP and is associated with a higher zinc-protoporphyrin fraction than EPP. EPP and XLPP patients are frequently subject to mild hypochromic anemia associated with iron deficiency. PPIX is eliminated by the liver. It may cause gallstones and cholestatic hepatitis. In approximatively 3% of cases, acute cholestasis progresses to liver failure.

In the bone marrow, heme synthesis is mainly controlled by intracellular labile iron by post-transcriptional regulation. Indeed, translation of *ALAS2* mRNA, the first and rate-limiting enzyme of the heme biosynthesis pathway in the erythroid tissue, is inhibited when iron availability is low [8,9]. Moreover, it has been shown that the expression of FECH, an iron-sulfur cluster enzyme, is regulated by intracellular iron level [10].

Accordingly, there is accumulating evidence that iron status can mitigate disease expression in patients with erythropoietic porphyrias. In CEP, the induction of an iron deficiency was able to drastically decrease hematological and cutaneous symptoms in some patients [11,12,13] as well as in cellular and mouse models [14]. There are conflicting reports on whether or not iron supplementation is beneficial in erythropoietic protoporphyrias [6,7,15,16,17,18,19,20,21,22,23,24,25,26].

In the review, we will firstly summarize the clinical data available on the influence of iron status on erythropoietic porphyrias symptoms. To illustrate practically how iron deficiency can limit the disease symptoms, we will describe data on one EPP and one CEP patient. Then, we will describe how iron availability can modulate the heme biosynthetic pathway activity, with a focus on ALAS2 and FECH regulation in patients with erythropoietic porphyrias. Finally, we will review FECH interactions with mitochondrial proteins involved in iron metabolism.

## 2. Iron Supplementation in Erythropoietic Porphyrias

### 2.1. Erythropoietic Protoporphyrias

As stated above, iron deficiency and hypochromic anemia is a frequent finding in protoporphyria. In a cohort of 226 North American patients (90.3% with EPP and 9.7% with XLPP), anemia was reported by 95 patients with EPP (46.6%) and by 12 patients with XLPP (54.5%) [7]. The diminished iron stores in EPP and XLPP seems not to be related to a lack of iron absorption or an inappropriate hepcidin regulation [27]. Such a finding may lead to the prescription of oral iron therapy in order to restore the hemoglobin level. Moreover, it can be hypothesized that iron supplementation could decrease PPIX level in erythropoietic protoporphyria patients by making FECH substrate more available and thus reducing the accumulation of PPIX.

Several attempts of iron substitution therapies have been reported and are summarized in Table 1. Interventions aiming to treat acute cholestasis were not recorded here, even so, some of them could be considered as an iron intake. Indeed, iterative transfusions and blood exchanges constitute a significant iron intake. However, they are usually prescribed to decrease PPIX production by reducing the patient’s hematopoiesis (and, for blood exchange, by removing the patient’s erythrocytes by PPIX-free ones). Repeated transfusions, when aiming to reduce photosensitivity, are also shown in Table 1 [28,29]. However, symptoms improvement could rather be related to hematopoiesis slowing than to iron intake.

Since the description of XLPP [6], several reports described iron therapy to be beneficial in XLPP [6,7,17]. Indeed, in XLPP, over activated ALAS2 is responsible for the accumulation of PPIX while FECH activity is conserved. In this case, iron is the limiting substrate and FECH is utilizing zinc (Zn) to form PP-Zn, biochemically distinguishing EPP from XLPP. It can be hypothesized that iron supplementation would decrease PPIX accumulation by rendering FECH substrate more available.

It cannot be excluded that some of the early reports, before molecular diagnosis could be obtained, are unknowingly describing XLPP patients instead of EPP, as they account for up to 10% of the patients in the USA [7].

When considering only cases where oral or intravenous iron were given to patients, excluding known XLPP patients, there are 11 reports in the literature [15,16,18,19,20,21,22,23,26,30]. Of those, eight are describing a worsening of symptoms under treatment [16,18,19,21,22,23,30] whereas three reported an improvement [15,20,26]. Of those three, one was not molecularly diagnosed [20] and iron was prescribed in an attempt to treat liver disease. The two other cases described the same patient [15,26]. He was diagnosed with EPP, with confirmed *FECH* mutations. His photosensitivity and overall well-being improved under oral iron therapy and, later, under repeated intravenous iron infusions. Curiously, under oral iron, his PPIX and hemoglobin (Hb) levels remained stable and, under i.v. iron, his PPIX decreased without a substantial change in Hb level.

Even if all of those 10 patients were EPP patients, the vast majority (80%) had a worsening of photosensitivity under iron treatment. This pleads for careful use of those medications. As already proposed by several authors, microcytic anemia in EPP patients should only be treated in case of a significant impact in patient quality of life and under close surveillance of patient’s PPIX, liver enzymes, Hb and ferritin level [16]. This recommendation is based on a small number of case reports, a significant fraction of which has not been characterized at the molecular level. In addition, patients were treated with different amounts of oral or i.v. iron. A definitive answer regarding the role of iron in erythropoietic protoporphyrias can only be provided by a randomized clinical trial.

Considering that iron therapy could increase PPIX concentration in EPP, and the fact that *ALAS2* mRNA translation is regulated by iron availability (see below), it could be hypothesized that inducing iron deficiency in EPP patients might decreaseALAS2 activity and thus, PPIX accumulation. In Figure 2, data on PPIX level and iron status of an EPP patient treated repeatedly with phlebotomies is presented. As frequently described, she had an Hb level close to the lower limit of normal (Hb 11.7 g/dL; N: 11.5–14.9), with a ferritin level of 10 µg/L (N: 8–252 µg/L). Her baseline total erythrocytes porphyrins was 84.1 µmol/L erythrocytes. Since spring 2018, at the age of 42, she was treated with iterative phlebotomies in small volumes (approximatively 3 mL/kg) to induce iron deficiency without too great a decrease in Hb level and to avoid an intense stimulation of hematopoiesis. Treatment frequency was adapted to patient tolerance and Hb level, which never fell below 8.5 g/dL. This procedure was repeated every spring from 2018 to 2021. Median total erythrocytes porphyrins decrease was 46.4% (min 44%, max 51%). More than the ferritin level, we observed that transferrin saturation correlated better with erythrocytes porphyrins level (panel a). When it decreased, we concomitantly observed an increase in PP-Zn fraction (panel b). This treatment succeeded in lessening the patient photosensitivity but it failed to fully normalize the PPIX accumulation.

### 2.2. Iron Deficiency in CEP

On the contrary, the influence of iron status on CEP pathology is much more clear-cut. Egan et al. reported the first case of dramatic improvement in photosensitivity and hemolysis in a CEP patient with iron deficiency caused by gastrointestinal bleeding [11]. Her symptoms worsened concomitantly with the gastrointestinal bleeding resolution. This was accompanied by a rise in ferritin, porphyrins in urines and in lactate dehydrogenase (LDH) levels. She was then treated with iron chelators, which induced a novel improvement in symptoms as well as a massive reduction of porphyrins levels and LDH normalization.

Based on this report, we and others [12,13] decided to treat patients with CEP without transfusion-dependent anemia by iterative phlebotomies. Phlebotomies were preferred to an iron chelator because of the important toxicity associated with the latter. Moreover, phlebotomy is accessible worldwide and is a simple and non-expensive procedure. This procedure resulted in a massive decrease (≈90%) in plasma and urines porphyrins. Porphyrins levels of patients treated by phlebotomies were similar to the residual concentrations seen in patients after hematopoietic stem cell transplantation. The rapid decrease in porphyrins occurred once patients were deeply iron deficient. Initially, a rise in porphyrins can be observed as patients’ hematopoiesis is stimulated in a context of still-available iron. A patient’s biological follow-up is presented in Figure 3. He was diagnosed at age 22 with a moderate form of CEP. He had a mild, asymptomatic, non-transfusion dependent, anemia (Hb = 13.1 g/dL; N: 13.4–16.7). He presented with moderate scarring and hypertrichosis of photoexposed areas. A homozygous mutation in *UROS* was present (c.660+4delA, already described in a CEP patient [31]) resulting in decreased UROS activity (2.5 U/mg Hb/h, N > 6). His baseline urinary porphyrins was 1523 mmol/mmol of creatinine (N < 30). One phlebotomy of 200–300 mL was performed every month or every 2 months. At first, porphyrins increased in urine and plasma, probably due to hematopoiesis stimulation. After two and a half months, porphyrins started to decrease. Seven months after the treatment initiation, urine porphyrins were completely normalized. The patient reported asthenia and Hb level was 9.3 g/dL. Phlebotomies were performed less frequently to maintain Hb level above 10 g/dL without a major increase in porphyrins levels. Since the beginning of treatment, there was no novel cutaneous manifestation.

## 3. Modulation of Heme Biosynthesis Pathway Activity by Iron Availability in the Erythron

Heme formation requires iron as it is incorporated into PPIX by FECH in the last step of the biosynthesis pathway. In the erythroid tissue, more than being only a substrate in heme synthesis, iron concentration is involved in the regulation of the whole pathway.

### 3.1. IRE/IRP System

Several mRNAs coding for proteins involved in iron metabolism include an iron responsive element (IRE) [9]. When iron availability is low, iron regulatory proteins (IRPs) 1 and 2 are able to bind to IRE [9,32]. If the IRE is in the 5′ UTR (ALAS2, ferroportin, L and H-ferritin) or in the 3′ UTR (RTf1, DMT1), of mRNA, IRP binding will lead to a translation inhibition or to an enhanced mRNA stability, respectively. Thus, when cellular iron is low, *ALAS2* mRNA translation will be inhibited and heme synthesis is repressed. Conversely, in iron replete conditions, IRPs lose their ability to bind the 5′ IRE either by degradation (IRP2) or iron-sulfur cluster (ISC) formation (IRP1). IRP1 then functions as a cytoplasmic aconitase.

### 3.2. Iron-Sulfur Proteins

Iron-sulfur proteins (Fe-S proteins) are involved in various metabolic reactions, such as redox reactions in respiratory complexes, citric acid cycle or DNA synthesis [33,34]. ISC biosynthesis takes place in the mitochondria and is coordinated by the ISC machinery [35]. ISCs have the ability to transfer single electrons. Two Fe-S proteins are involved in the heme biosynthesis pathway. First, IRP1, a cytosolic counterpart to mitochondrial aconitase, which catalyzes the isomerization of citrate to isocitrate [36]. An increase in intracellular iron induces the formation of an ISC, preventing IRP1 binding to IREs.

Second, the last enzyme of the heme biosynthesis pathway, FECH, is also an Fe-S protein. The ISCs do not participate in the catalytic activity [37]. However, mutations altering one of the four cysteine residues involved in the cluster fixation resulted in a decrease in FECH activity [38,39] and in a typical EPP phenotype in patients [40]. It has been shown that FECH expression is regulated by iron availability [10]. Indeed, in human K562 cells cultivated under iron-depleted conditions, FECH activity and protein level were decreased with a conserved amount of mRNA. Bacterial FECH, lacking the ISC, did not show the same decrease in FECH activity. Thus, FECH is regulated by intracellular iron level, probably through its ISC. Post-transcriptional regulation of FECH was confirmed in vivo in *Irp2-/-* mice [41]. Furthermore, the stability of newly formed ferrochelatase protein was drastically impaired in murine erythroleukemia cells cultivated with an iron chelator [41].

### 3.3. FECH and ALAS2 Regulation by Iron in EPP Patients

Taken collectively, this data on iron regulation of ALAS2 and FECH suggests that when iron availability is low, there is an inhibition of *ALAS2* mRNA translation, combined with an increase in ferrochelatase protein degradation; all of it converging to decrease heme production. In patients with functional heme biosynthesis pathway, a chronic depletion in iron stores leads to microcytic hypochromic anemia with a moderate increase in PP-Zn, which highlights the partially conserved ferrochelatase activity even if iron availability is scarce. In EPP patients with deficient FECH, iron deficiency might lead to less PPIX accumulation by inhibiting ALAS2 but should also increase PPIX level by inhibiting FECH action. Therefore, the outcome on PPIX level might depend on the relative sensitivity of ALAS2 and FECH to iron deficiency. ALAS2 inhibition could sufficiently slow the heme pathway activity and prevent PPIX accumulation despite FECH inhibition and increased degradation. Conversely, if FECH inhibition, in iron deplete conditions, prevails over ALAS2 inhibition, there should be an increase in PPIX levels. In EPP patients with induced iron deficiency, the 45% decrease in PPIX level (vs. more than 90% decrease in uroporphyrin level in CEP patients) illustrates the dual effect of iron depletion.

Several studies of FECH and ALAS2 expression in EPP patients have been reported. The ALAS2 protein was shown to be more expressed in young erythrocytes from a small group of patients compared to control subjects [16]. *ALAS2* mRNA was also more abundant in patients. One could have expected lower protein level in patients compared to control subjects, given that under iron deficient state, which is frequent in EPP, *ALAS2* mRNA translation is inhibited. In this study, the increase in ALAS2 protein was less marked than that in mRNA, which could indicate a partial translation inhibition by the IRE/IRP system. In erythroleukemic K562 cell, FECH inhibition by *N*-methylprotoporphyrin resulted in marked increase in *ALAS2* mRNA without an increase in FECH expression [16]. Other studies by the same group showed that *ALAS2* mRNA was increased in K562 cells treated with an iron chelator in a dose-dependent manner [42]. This was not the case for *FECH*, causing the ratio of *ALAS2* mRNA on *FECH* mRNA to increase with the dosage. However, the abundance of aberrant *FECH* transcripts was increased in iron depleted cells [43]. This was associated with a decrease in total FECH protein as well as in ALAS2 (as expected due to the IRE/IRP translation inhibition).

Overall, this supports the idea that FECH deficiency increases ALAS2 expression by a yet unknown mechanism. It was proposed that in EPP patients, FECH deficiency leads to an ALAS2 overexpression which contributes to PPIX accumulation [42]. Thus, it is appealing to hypothesize that the frequent iron deficiency in patients mitigates the ALAS2 overexpression via the IRE/IRP system. Iron substitution could then further aggravate PPIX accumulation by lifting the inhibition on *ALAS2* mRNA translation.

The role of iron in mitigating erythropoietic porphyrias is further underlined by the description of a patient with erythropoietic protoporphyria caused by combination of mutations in *CLPX* and in the IRE domain of *ALAS2* [44]. The first mutation caused CLPX proteolytic activity to decrease and led to ALAS2 accumulation. The second prevents IRPs binding to *ALAS2* mRNA, thus impairing post-transcriptional inhibition under iron deficiency. Family members without the mutation in *ALAS2* accumulated moderate amount of PPIX and showed mild photosensitivity whereas the patient had highly elevated erythrocytes PPIX and symptoms as seen in EPP. Oral iron therapy was able to decrease PPIX level.

In CEP, it could be hypothesized that without the induction of ALAS2 expression, such as seen in FECH deficient patient, iron deficiency efficiently succeeds in inhibiting *ALAS2* mRNA translation via the IRE/IRP system. This would explain why inducing iron deficiency in the CEP patient is able to more successfully decrease the heme pathway intermediates accumulation. Following the same logic, an *ALAS2* gain-of-function mutation was shown to be responsible for a more severe CEP phenotype [45].

The *ALAS2* overexpression or gain-of-function mutations in erythropoietic porphyrias suggest that ALAS2 expression could be a target for therapeutics aiming to reduce its activity. Such a strategy has been used in acute hepatic porphyrias. Indeed, patients treated with givosiran, an inhibitor of ALAS1, the first enzyme of the heme biosynthesis pathway in the liver, showed sustained decreased in *ALAS1* mRNA and urinary toxic precursors of heme, delta aminolevulinic acid and porphobilinogen [46,47,48].

## 4. FECH Regulation at the Protein Level

Recent studies of FECH activity revealed that the enzyme is not only the final enzyme of the heme synthesis, but also an important regulator of the whole pathway [49]. For instance, heme production during erythropoiesis is regulated by EPO signaling through FECH phosphorylation, which induces an upregulation of its activity [50]. It has been shown that FECH works in interaction with several proteins involved in iron metabolism including mitoferrin-1 (MFRN1) [51], ABCB7 [52,53,54], ABCB10 [53,54,55], Frataxin [56], FAM210B [57] as well as other enzymes of the heme biosynthesis pathway, ALAS2 and protoporphyrinogen oxidase PPOX, the penultimate enzyme of the pathway [54].

As stressed in Medlock et al., 2015, a multi-protein complex made up of terminal enzymes of heme biosynthesis involved in iron metabolism would facilitate coordination of heme synthesis and iron uptake by the mitochondrion [51,54]. Moreover, it would protect the cell from reactive molecules, such as porphyrins, iron and heme.

MFRN1 is an iron importer localized at the inner membrane of the mitochondrion. It is encoded by the *SLC25A37* gene. MFRN1 was found in an oligomeric complex with FECH and ABCB10 [55]. ABCB10 is thought to stabilize MFRN1 [55]. The complex facilitates MFRN1-imported iron transfer to FECH for heme biosynthesis [55]. As stated before, FECH stability is decreased when cellular iron is depleted and less available for ISC formation [10]. In patients with reduced FECH activity, *SLC25A37* mRNA levels were reduced. Reduced MFRN1 could impair ion uptake by the mitochondrion leading to lesser ISC formation and to reduced FECH activity [58]. Reduced *SLC25A37* mRNA was found in patients with classical EPP, individuals with XLP and individuals with consistent biochemical studies for EPP without identifiable mutation in *FECH* or *ALAS2* [59].

ABCB7 was found to interact directly with FECH [52]. In cell cultures, *ABCB7* mRNA degradation led to the inhibition of heme biosynthesis and its overexpression led to an increase in heme content. It is thought that ABCB7 plays a role in the formation and maintenance of FECH ISC. In another study, a complex consisting of ABCB10, ABCB7 and FECH was isolated [53]. *ABCB7* knockdown cells showed loss of multiple Fe-S mitochondrial enzymes, increased mitochondrial iron associated with a defect in heme biosynthesis, IRP2 activation and MFRN1 upregulation. Consistently, *ABCB7* mutations can result in X-linked sideroblastic anemia with ataxia.

FAM210B is a mitochondrial inner membrane protein required for heme synthesis in differentiating erythroid cells [57]. Indeed, it facilitates iron import by MFRN1 to support sustained synthesis of heme and ISC during erythroid maturation. FAM210B deficient cells showed impaired ISC biosynthesis resulting in increased *ALAS2* mRNA translation inhibition by IRP1. In addition, it was proposed that FAM210B in an oligomer with FECH and PPOX, could facilitate the transfer of protoporphyrinogen to FECH [57]. Moreover, FAM210B seems to be required for FECH full activation. Thus, FAM210B connects the iron uptake by the mitochondrion and heme synthesis.

Frataxin-defective organisms accumulate iron in the mitochondrion and show deficient ISC biogenesis [35,60]. In the cytosol, there is a relative decrease in iron, resulting in activation of IRPs. It was demonstrated that frataxin is an iron-binding protein. During the ISC biosynthesis, frataxin could have the role of an iron donor or at least a regulatory role. Since frataxin binds to yeast and human FECH [61,62], it was proposed that it could provide iron to FECH [56,63].

Altogether, this indicates that FECH activity is not only regulated by iron availability but probably also by complex interactions with multiple protein partners, some of them involved in mitochondria iron metabolism. FECH seems to be situated at a crossroads between heme synthesis and ISC biosynthesis.

## 5. Conclusions

Iron plays a major role in heme biosynthesis as a substrate but also as a key regulator at several levels. Iron deficiency is probably protective in EPP and CEP, whereas its supplementation should be carried out with care, especially with regard to the hepatic complications of the disease in EPP. We propose that it should be restricted to symptomatic iron deficiency. On the contrary, XLPP patients seem to benefit from oral iron therapy. A few CEP patients were treated with success by iron depletion (by iron chelators or iterative phlebotomies). This treatment should be considered either in patients without transfusion dependent anemia, in patients for whom bone marrow allograft is not an option or both.

The regulation of heme synthesis by iron occurs at least three different levels. At the post-transcriptional level by the IRE/IRP system, through ISC formation and at the protein level via FECH interactions with proteins involved in iron metabolism. Although the role of iron in regulating heme synthesis and as a modifier of severity in erythropoietic porphyria is now strongly demonstrated, many questions remain to be elucidated. For example, there is still no explanation on why patients with erythropoietic protoporphyria are subject to iron deficiency. A better understanding of erythropoietic porphyrias pathogenesis might suggest therapeutics targets.

Data on patients with erythropoietic porphyrias suggests that ALAS2 is one of them. By slowing down the erythroid heme biosynthetic pathway, it could be expected to diminish the toxic intermediates accumulation. Recently, GlyT1, a glycine transporter of erythropoietic cells, was identified as a potential target. Glycine is a substrate of the first and rate limiting enzyme of heme biosynthesis, ALAS2. GlyT1 can be selectively inhibited by bitopertin. GlyT1 inhibition in rats results in a microcytic hypochromic anemia without iron-overload [64]. A mouse-model of beta-thalassemia treated with bitopertin showed improved hematological parameters with diminished ineffective erythropoiesis, but it failed to improve erythropoiesis in patients [65,66]. A phase II trial to evaluate bitopertin efficacy in decreasing porphyrins accumulation in red blood cells of patients with erythropoietic porphyrias is scheduled in 2022.

## Figures and Tables

**Figure 1 metabolites-11-00798-f001:**
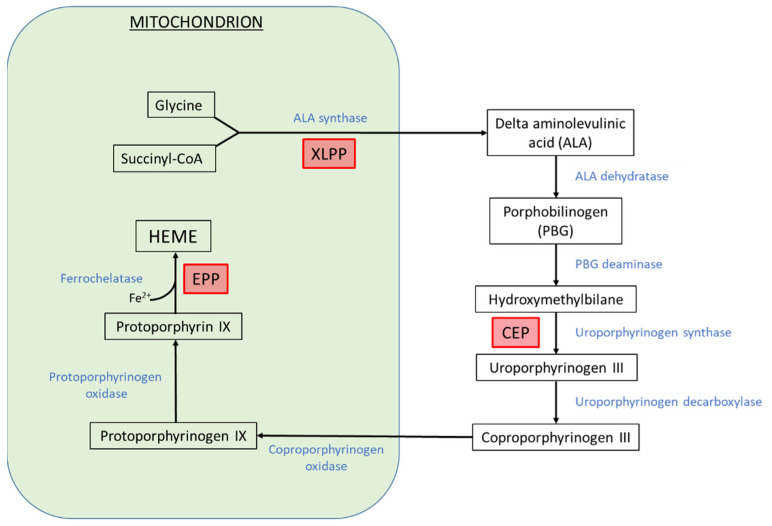
Heme biosynthesis pathway. Enzymes are indicated in blue. Erythropoietic porphyrias are indicated in red boxes in front of the corresponding enzyme dysfunction. CEP: congenital erythropoietic porphyria; EPP: erythropoietic protoporphyria; XLPP: X-linked protoporphyria.

**Figure 2 metabolites-11-00798-f002:**
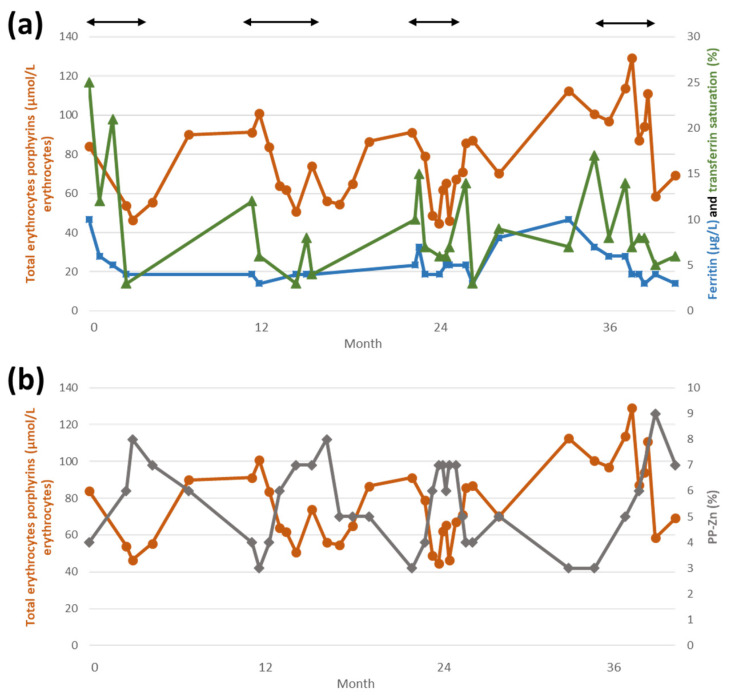
EPP patient’s biological parameters. Black arrows indicate treatment periods. (**a**) Total erythrocytes porphyrins (µmol/L erythrocytes, dots), ferritin (µg/L, squares) and transferrin saturation (%, triangles); (**b**) total erythrocytes porphyrins (µmol/L erythrocytes, dots) and PP-Zn (protoporphyrin-Zn, %, diamonds). EPP: erythropoietic protoporphyria.

**Figure 3 metabolites-11-00798-f003:**
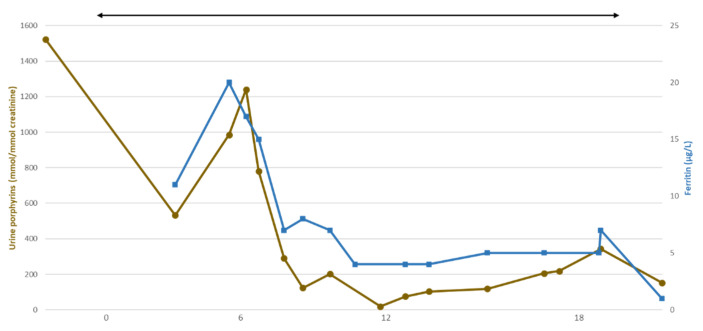
CEP patient’s biological parameters. Black arrow indicates iterative phlebotomies every 1 or 2 months. Total urine porphyrins (mmol/mmol creatinine, dots), ferritin (µg/L, squares). CEP: congenital erythropoietic porphyria.

**Table 1 metabolites-11-00798-t001:** Patients with erythropoietic protoporphyria treated with iron.

Authors	Year	Number of Patients	Sex	Type	Molecular Diagnosis	Biochemical Diagnosis	Intervention	Clinical Outcome	Biochemical Outcome
Reed et al.	1970	1	F	NA	no	yes	oral iron therapy	worsening	NA
Baker et al.	1971	1	F	NA	no	yes	oral iron therapy	worsening	PPIX increase
Bechtel et al.	1981	1	M	NA	no	yes	repeated transfusions	improvement	PPIX decrease
Dobozy et al.	1983	5	1 F and 4 M	NA	no	yes	repeated transfusions	improvement	PPIX decrease
Graham-Brown et al.	1984	1	F	NA	no	yes	oral iron therapy	worsening	PPIX decrease after iron therapy discontinuation
Gordeuk et al.	1986	1	F	NA	no	yes	oral iron therapy *	NA	PPIX decrease
Milligan et al.	1988	2 **	F	NA	no	yes	oral iron therapy	worsening	PPIX increase
McClements et al.	1990	1	F	NA	no	yes	oral iron therapy	worsening	PPIX decrease after iron therapy discontinuation
Todd et al.	1992	1	M	NA	no	yes	repeated transfusions	worsening	PPIX increase
Holme et al.	2007	1	M	EPP	yes	yes	oral iron therapy	improvement	stable PPIX
Whatley et al.	2008	1	M	XLPP	yes	yes	oral iron therapy	improvement	PPIX decrease
Whalin et al.	2011	1	F	EPP	yes	yes	oral iron therapy	worsening	stable PPIX
Bentley et al.	2013	1 ***	M	EPP	yes	yes	IV iron therapy	improvement	PPIX decrease
Barman-Aksözen et al.	2015	2	F	EPP	yes	yes	IV or oral iron therapy	worsening	PPIX increase (one patient) stable PPIX (one patient)
Balwani et al.	2017	8	F	XLPP	yes	yes	oral iron therapy	improvement (7/8)	NA

* Indication: hepatic dysfunction; ** 4 patients are reported but 2 were already described by [30] and [21]; *** same patient as in [26]; PPIX: protoporphyrin IX; F: female; M: male; NA: non-available; EPP: erythropoietic protoporphyria; XLPP: X-linked protoporphyria.

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
