# Peer review of "Iron, Heme Synthesis and Erythropoietic Porphyrias: A Complex Interplay"

_metabolites, 2021, doi:10.3390/metabo11120798_

Round 1

Reviewer 1 Report

In this manuscript, Poli and collaborators summarizes the role of the iron in the regulation of the heme synthesis and its importance on the treatment of Congenital Erythropoietic Porphyria, Erythropoietic Protoporphyria and X-linked Dominant Protoporphyria. The manuscript is concise and completely supported by the refereces used. My congratulations for the authors, as I could not find any uncorrect and non supported information.

As a minor suggestion:

1- English language should be reviewed as I read a couple of incorrect words.

2- I would not use the exact date on the Figure 1 and 2, as it looks a little bit weird. I would use a time axis (1 month, 2 months, 3 months...).

Again, congratulations for the well done work!!!

Author Response

The authors thank the reviewer for his evaluation of the manuscript. Please find below our response to the first reviewer’s comments

Point 1: English language should be reviewed as I read a couple of incorrect words.

Response 1: The authors once again revised the manuscript. We corrected several misspelling errors or incorrect wording, including those pointed out by reviewer 2.

Point 2: I would not use the exact date on the Figure 1 and 2, as it looks a little bit weird. I would use a time axis (1 month, 2 months, 3 months...).

Response 2: We modified the figures as suggested by the reviewer. In order not to overload the figures, we chose to represent time intervals of six months or one year.

Reviewer 2 Report

The review “Iron, heme synthesis and erythropoietic porphyrias: a complex interplay” by Antoine Poli et al, is an informative and interesting one. The authors have succinctly provided many key facts regarding the involvement of iron and the two key enzymes of the heme biosynthesis pathway: ALAS2 and FECH in the regulation of erythropoiesis and in the pathogenesis of erythropoietic porphyrias. Erythropoiesis is a complex process regulated by various signalling pathways and a host of lineage-specific transcription factors that regulate erythropoiesis-specific gene expression, which includes the enzymes responsible for heme biosynthesis. The key role of the heme biosynthesis pathway in preventing anemia and erythropoietic porphyrias is well established. In this review, the authors have taken in to consideration the critical role of the level of iron in determining the ALAS2 and FECH activity, and how it is affecting erythropoiesis, in analysing patients suffering from erythropoietic porphyrias. However, the authors need to update the review with more patient’s data with molecularly characterized erythropoietic porphyrias and supporting literature to make a clear conclusion and delineate their outlook for the mode of iron therapy as a definitive therapeutic intervention. The patients data presented are not enough and show conflicting outcome following iron therapy.

Comments:

  1. The authors need to give a brief description of heme biosynthesis pathway with the enzymes involved in each step, perhaps a sketch diagram indicating the step leading to erythropoietic porphyrias will be very helpful to the readers, who are uninitiated to the area of erythropoiesis but will find it interesting to read.
  2. Although it gives a comprehensive idea about the mechanisms of erythropoietic porphyrias pathogenesis, the main drawback is the relatively small number of patient data cited in the review.
  3. Out of the patients data provided, some are not clear about erythropoietic porphyrias due to lack of molecular characterization. Moreover, from the studies cited, conflicting role of iron treatment and the routes of iron treatment on protoporphyrin IX (PPIX) levels have been reported. Hence, a clear understanding of iron levels on the regulation of these key enzymes in erythropoietic porphyrias is still lacking.
  4. The authors need to incorporate more patient data with clear molecular and biochemical characterization of erythropoietic porphyrias to derive a solid conclusion on involvement of iron in the pathogenesis and cure of this disorder.
  5. The English needs to be improved throughout.
  6. At lines, 13 and 21, erythropoïetic should be erythropoietic.
  7. Line 44 should be –subdivided in to two types depending on…..
  8. Line 53-XLPP is…..
  9. Line 66- as well as in cellular and mouse models.
  10. Line 85- erythropoietic protoporphyria patients by making Fech substrate…..
  11. Line 89-90 is not a complete sentence (Indeed, iterative transfusions 89 and blood exchange are significant iron intake.)
  12. Line 91- patient’s hematopoiesis
  13. Line 94- rather be related to slowing of hematopoiesis…….
  14. Line 107- When considering only cases, where oral or intravenous…..
  15. Line 108- there are 11 reports in the literature.
  16. Line 110- whereas three reported an improvement.
  17. Line 117- stable and, under i.v. iron, his……
  18. Line 135- deficiency without too much decrease in Hb level……
  19. Line 143- succeeded in lessening the patient’s photosensitivity…….
  20. Line 146- the influence of iron status on CEP pathology is much…..
  21. Line 151- a rise in ferritin, porphyrins in urines and in LDH levels. Also, expand LDH.
  22. Line 170- A patient’s biological follow-up is pre……
  23. Line 174- A homozygous mutation in UROS was…. Expand UROS.
  24. Line 175- CEP patient [31]) resulting in decreased……
  25. Line 179- After two and a half months…….
  26. Line 184- Since the beginning of treatment, there…………..
  27. Lines 199-201 - If the IRE is in the 5’ UTR (ALAS2, ferroportine, L and H-ferritin) or in the 3’ UTR (RTf1, DMT1), of mRNA……..
  28. Line 226- confirmed in vivo in Irp2-/- mice [41].
  29. Line 248- Several studies of FECH and ALAS2 expression in EPP patients have been reported.
  30. Line 260- FECH, causing the ratio of………..
  31. Line 295- of heme, ALA and PBG [47–49]. Expand ALA and PBG.
